# Left Ventricular Lead Placement Guided by Reduction in QRS Area

**DOI:** 10.3390/jcm10245935

**Published:** 2021-12-17

**Authors:** Mohammed Ali Ghossein, Francesco Zanon, Floor Salden, Antonius van Stipdonk, Lina Marcantoni, Elien Engels, Justin Luermans, Sjoerd Westra, Frits Prinzen, Kevin Vernooy

**Affiliations:** 1Cardiovascular Research Institute Maastricht (CARIM), Maastricht University, 6229 ER Maastricht, The Netherlands; frits.prinzen@maastrichtuniversity.nl; 2Santa Maria Della Misericordia General Hospital, 45100 Rovigo, Italy; franc.zanon@iol.it (F.Z.); lina.marcantoni@gmail.com (L.M.); 3Maastricht University Medical Center, Department of Cardiology, Maastricht University, 6229 HX Maastricht, The Netherlands; floor.salden@mumc.nl (F.S.); twan.van.stipdonk@mumc.nl (A.v.S.); justin.luermans@mumc.nl (J.L.); kevin.vernooy@mumc.nl (K.V.); 4Yale New Haven Hospital, New Haven, CT 06510, USA; elienengels@gmail.com; 5Radboud University Medical Center, Radboud University Nijmegen, 6525 GA Nijmegen, The Netherlands; sjoerd.westra@radboudumc.nl

**Keywords:** cardiac resynchronization therapy, LV lead placement, QRS area, invasive hemodynamic measurements, heart failure

## Abstract

*Background*: Reduction in QRS area after cardiac resynchronization therapy (CRT) is associated with improved long-term clinical outcome. The aim of this study was to investigate whether the reduction in QRS area is associated with hemodynamic improvement by pacing different LV sites and can be used to guide LV lead placement. *Methods*: Patients with a class Ia/IIa CRT indication were prospectively included from three hospitals. Acute hemodynamic response was assessed as the relative change in maximum rate of rise of left ventricular (LV) pressure (%∆LVdP/dt_max_). Change in QRS area (∆QRS area), in QRS duration (∆QRS duration), and %∆LVdP/dt_max_ were studied in relation to different LV pacing locations within a patient. *Results*: Data from 52 patients paced at 188 different LV pacing sites were investigated. Lateral LV pacing resulted in a larger %∆LVdP/dt_max_ than anterior or posterior pacing (*p* = 0.0007). A similar trend was found for ∆QRS area (*p* = 0.001) but not for ∆QRS duration (*p* = 0.23). Pacing from the proximal electrode pair resulted in a larger %∆LVdP/dt_max_ (*p* = 0.004), and ∆QRS area (*p* = 0.003) but not ∆QRS duration (*p* = 0.77). Within patients, correlation between ∆QRS area and %∆LVdP/dt_max_ was 0.76 (median, IQR 0.35; 0,89). *Conclusion*: Within patients, ∆QRS area is associated with %∆LVdP/dt_max_ at different LV pacing locations. Therefore, QRS area, which is an easily, noninvasively obtainable and objective parameter, may be useful to guide LV lead placement in CRT.

## 1. Introduction

Cardiac resynchronization therapy (CRT) has become one of the most successful treatments for heart failure. Current guidelines indicate a CRT device in patients with a reduced left ventricular (LV) systolic function in combination with electrocardiographic signs of intraventricular conduction abnormalities, presented as a wide QRS duration and the presence of an left bundle branck block (LBBB) QRS morphology [1,2].

The electrical substrate amenable to CRT is hypothesized to be delayed electrical activation of the left ventricular (LV) lateral wall, which can be measured by invasive electrophysiological examination. Vectorcardiographically determined QRS area is able to detect the electrical substrate better than QRS duration or LBBB morphology [3]. Moreover, baseline QRS area predicts echocardiographic response after CRT, and is strongly associated with clinical outcomes, including mortality [4,5,6,7].

In addition to baseline QRS area of the intrinsic rhythm, it was recently demonstrated that also reduction in QRS area after CRT is of clinical importance. A larger reduction in QRS area was associated with echocardiographic and clinical response after CRT [6,7,8]. Although these data were obtained using single measurements per patient, the results raised the question whether reduction in QRS area might be used to guide LV lead positioning in the individual patient.

Currently, LV lead positioning is guided by coronary venous anatomy. However, several studies were not able to show a significant difference in outcome between various anatomical lead locations [9]. At least part of these negative results may be explained by differences between patients, pointing to the need of personalization of LV lead placement. To this purpose, several approaches have been proposed, such as measurement of latest electrical (Q-LV time; ECG imaging) or mechanical (speckle tracking echocardiography, cardiac magnetic resonance) activated region, all or not in combination with localizing scar [10,11,12,13,14].

However, these studies use the baseline condition and do not take into account the change in activation caused by CRT. The use of the change in QRS area may have two advantages: taking into account the change in activation and the fact that this variable is easily measured. Therefore, the present study aims to investigate whether the reduction in QRS area can be used to guide LV lead placement. To this purpose, the acute hemodynamic changes during various LV lead positions per patient were studied in relation to the change in QRS area during CRT implantation. 

## 2. Methods

### 2.1. Study Population

We conducted an analysis on prospectively included CRT patients from three hospitals with a class Ia or class IIa indication for CRT. The analyses were performed on the patients that were included in two previously conducted studies evaluating the acute hemodynamic effects of CRT [15,16]. All patients underwent implantation of a CRT device with a quadripolar LV lead. In the study of Zanon et al. [16], twenty-nine consecutive patients from one center (Santa Maria della Misericordia General Hospital, Rovigo, Italy) referred for CRT were included, and in the study of Salden et al. [15], twenty-seven CRT patients from two additional hospitals (Maastricht University Medical Center+ and Radboud University Medical Center, Nijmegen, The Netherlands) were included. Both studies were previously approved by their local ethics committee and were conducted in compliance with the Declaration of Helsinki.

### 2.2. Study Protocol and Hemodynamic Measurements

In both studies, patients underwent CRT implantation with the right ventricular lead in the (apical) septum, and the atrial lead in the right atrial appendage. 

In the study of Zanon et al., a telescopic approach was used for cannulation of the coronary sinus and sub cannulation of all suitable collateral veins, which allows continuous selective navigation with angiographic visualization [17]. All available veins were cannulated and then targeted with an LV quadripolar pacing lead. Paced 12-lead ECG tracings were recorded from every patient and relative change in maximum rate of rise of left ventricular (LV) pressure (%∆LVdP/dt_max_) measurements were performed with a PressureWire Certus and PhysionMon software (St. Jude Medical Systems AB, Uppsala, Sweden). For these LVdP/dtmax measurements a 4F multipurpose catheter was inserted through a femoral or radial arterial access site and placed in a stable LV position. The AV- and VV-delays were fixed at 130 ms and 0 ms, respectively. 

In the study from Salden et al., the LV lead was positioned in a suitable coronary vein on the posterolateral LV wall. In these patients, biventricular pacing measurements from a quadripolar LV lead were performed with a proximal and distal pacing configuration. At each configuration, 12-lead ECG and LVdP/dt_max_ measurements were taken, with the LV lead in posterolateral position. AV-delay was set 60 ms shorter than the intrinsic PR-interval, with a maximum AV-delay of 150 ms, and the VV-delay was set at 0 ms. For the LVdP/dt_max_ measurements a 0.014-inch PressureWire X (Abbott, St Paul, Minnesota, MN, USA) was introduced via a 4F multipurpose catheter through a femoral artery and placed in a stable LV position. 

In both studies, the average LVdP/dt_max_ of at least 10 beats at baseline (AAI pacing) and during pacing from each LV lead position was measured in each patient. Ventricular extrasystoles were excluded. The %∆LVdP/dt_max_ was calculated as the percentual change of LVdP/dt_max_ with every pacing site compared to baseline. 

### 2.3. Electrocardiographic Data

ECGs at baseline and per LV pacing location were digitally stored for offline analysis [8]. For QRS area calculations, 12-lead ECG signals were extracted from digitally stored PDF files and converted into three orthogonal VCG (X-, Y-, and Z-) leads using the Kors conversion matrix in the custom made Matlab software (Mathworks Inc.) as previously described [8]. QRS area was calculated as (X_area_^2^ + Y_area_^2^ + Z_area_^2^)^1/2^ (Figure 1). QRS duration was determined using the same software. To quantify the degree of resynchronization, the absolute difference in both QRS area and QRS duration between baseline and biventricular pacing (∆QRS area and ∆QRS duration) were calculated. QRS morphology was defined based on ESC 2013 criteria [18]. 

### 2.4. Statistical Analysis

Statistical analysis was carried out using the IBM SPSS package, version 26 (SPSS Inc., Chicago, IL, USA) and Graphpad Prism version 9.2.0 for Windows (GraphPad Software Inc., San Diego, CA, USA). Continuous and categorical variables are reported as mean ± SD and counts (percentages), respectively. Continuous variables were compared within patients using a paired *T*-test. A one-way ANOVA or mixed-effects model was used when comparing more than two categories. To correct for the within-patient measurements, a Geisser–Greenhouse correction was applied. Pearson’s and Spearman’s correlation coefficient were calculated to investigate the relation between ∆QRS area and %∆LVdP/dt_max_. 

## 3. Results

### 3.1. Baseline Characteristics

In 52 patients (Figure 2), ECGs and LVdP/dt_max_ data were available for analysis. This cohort represented a general CRT population with an age of 70 ± 9 years, of which 79% was male, and 50% had an ischemic cardiomyopathy. All patients were in NYHA functional class II or III, with a left ventricular ejection fraction (LVEF) of 29 ± 9%. Sinus rhythm was present in 71% of the patients during implantation with an average QRS duration of 169 ± 18 ms and QRS area of 108 ± 44 µVs (Table 1). LBBB was present in 67%, non-specific intraventricular conduction delay (IVCD) in 12%, right bundle branch block (RBBB) in 8%, and RV pacing in 13% of the studied patients.

### 3.2. ∆QRS Area, ∆QRS Duration, and %∆LVdP/dt_max_ at Various LV Lead Positions

Information on ∆QRS area, ∆QRS duration, and %∆LVdP/dt_max_ during biventricular pacing at different anatomical LV lead positions (anterior, anterolateral, lateral, posterolateral, posterior) was available in 26 patients (Figure 1). A mixed-effects statistical model showed that both QRS area and %LVdP/dt_max_ changed significantly within each patient, both dependent on the LV pacing site (*p* = 0.001 and *p* = 0.0007, respectively, Figure 3A,C, left). ∆QRS area was significantly larger with anterolateral (−43 µVs) as compared to posterolateral (−17 µVs), and posterior (+9 µVs) LV pacing, and it was significantly larger with lateral (−48 µVs) LV pacing as compared to anterior (−13 µVs) and posterior LV pacing (Figure 3A). The results on the %∆LVdP/dt_max_ at various LV lead positions were in concordance with the results on ∆QRS area. The largest %∆LVdP/dt_max_ was found during LV lateral wall pacing (24%), with a significantly higher increase than LV anterior (12%) or posterior (11%) wall pacing (Figure 3A, right). There was no significant difference in ∆QRS duration between the different pacing locations (Figure 3B, left).

### 3.3. Relationship between Anatomical LV Lead Position and Largest QRS Area Reduction and Relative LVdP/dt_max_ Increase

In 21 patients with at least two different anatomical LV lead pacing locations, the largest ∆QRS area was achieved from the lateral pacing position in ten patients (48%), from the anterolateral position in five patients (24%), from the posterolateral wall in four patients (19%), from the anterior position in two patients (1%), and from the posterior position in one patient (0.5%). In these patients, the largest LVdP/dt_max_ increase was achieved from the lateral pacing position in eleven patients (52%), from the anterolateral position in six patients (29%), from the posterolateral position in four patients (19%), and from the posterior position in one patient (0.5%).

### 3.4. ∆QRS Area and %∆LVdP/dt_max_ with Pacing from the Distal and Proximal LV Lead

From 43 patients (Figure 1), ∆QRS area and relative ∆LVdP/dt_max_ could be studied with LV pacing from the proximal and distal electrode pair on the quadripolar LV lead. The ∆QRS area was significantly larger with pacing from the proximal LV electrode as compared to the distal electrode (−39μVs vs. −17μVs, *p* = 0.0005, Figure 3A, right). In addition, ∆LVdP/dt_max_ was significantly larger when pacing from the proximal LV electrodes, although the difference was smaller than for ∆QRS area (18 vs. 16%, *p* = 0.0063, Figure 3C, right). There was no significant difference in ∆QRS duration when comparing LV pacing on the proximal versus distal electrodes (Figure 3B, right). 

### 3.5. Relation between ∆QRS Area and ∆LVdP/dt_max_

In 188 different pacing measurements, a Spearman’s correlation test found a significant association between ∆QRS area and ∆LVdP/dt_max_ (R_s_ = 0.47, *p* < 0.0001). In 36 out of the 43 patients (84%) with at least two different pacing measurements, %∆LVdP/dt_max_ was within 5% of the largest %∆LVdP/dt_max_ of that patient. Within the 21 patients with at least three different LV pacing sites, the correlation between ∆QRS area and %∆LVdP/dt_max_ was determined. As shown in Figure 4, this relation varied between patients, as shown by the variation in slope and intercept between patients. However, for the vast majority of patients, correlation between ∆QRS area and %∆LVdP/dt_max_ was strong (median R = 0.76, interquartile range 0.35–0.89, Figure 4).

## 4. Discussion

### 4.1. Main Findings

The present study in patients with a CRT indication shows that the reduction in QRS area during biventricular pacing correlates well with acute hemodynamic improvement. LV lateral and anterolateral pacing reduced QRS area most along with maximal hemodynamic improvement. In addition, when pacing with quadripolar LV leads, pacing proximally showed the largest reduction in QRS area and largest increase in LVdP/dt_max_. Such relations were not observed for the change in QRS duration. Moreover, within patients a highly significant relation between ∆QRS area and %∆LVdP/dt_max_ was found. These results indicate that the reduction in QRS area during biventricular pacing can be used to guide the LV lead placement in heart failure patients undergoing CRT-device implantation. 

### 4.2. QRS Area Reduction Represents Resynchronization Better Than QRS Duration or Morphology Change

This study is the first to investigate the relationship between ∆QRS area and acute hemodynamic improvement within CRT patients. The results of the current analysis support previous findings in which ∆QRS area was studied in CRT registries in relation to clinical and echocardiographic outcomes. Okafor et al. found that a reduction in QRS area in combination with a reduction in QRS duration was significantly associated with long-term cardiac and total mortality, major adverse cardiac events, and ventricular arrhythmias [6]. More recently, a study from our institute showed that in a large cohort of 1299 patients, ∆QRS area was independently associated with long-term survival and echocardiographic response [8]. The current analysis shows that ∆QRS area also correlates with acute hemodynamic improvement during biventricular pacing at various LV pacing sites.

Our findings were also in agreement with the study of De Pooter et al., in which ΔQRS area and ΔQRS duration were measured in different pacing configurations and were studied in relation to acute hemodynamic response. In the study of De Pooter et al., both reduction in QRS area and QRS duration were predictive of acute hemodynamic response. Nevertheless, QRS area reduction was the best predictor [19]. More studies have investigated ∆QRS duration in relation to clinical and echocardiographic outcomes. This has, however, led to mixed results. A sub study of the PROSPECT trial showed an association between QRS duration reduction and the combined endpoint of echocardiographic response and clinical improvement [20]. In the REVERSE trial, however, this association was not observed [21]. In the present study, we found that QRS duration does not change significantly with different pacing locations within a patient and could not be used to guide LV lead placement.

A probably important factor for the conflicting ∆QRS duration results is the inter- and intra-observer variability in measuring QRS duration during biventricular pacing. As shown by de Pooter et al., the variability of these measurements amounted to around 20 ms, and the technique used for QRS duration measurement even influenced the association with response to CRT [22,23]. These findings suggest that an error margin for measuring QRS duration exists, and that this error margin could have consequences for the prediction of response to CRT. For QRS area, the variability is less than half of that of QRS duration [22], making QRS area a more accurate and objective measure. 

The QRS area measurement combines both QRS duration and morphology. The importance of both QRS duration and morphology change has been demonstrated by Sweeney et al. in which the combination of a biventricular paced QRS fusion pattern and QRS duration reduction ≥25 ms was associated with echocardiographic response to CRT [24]. Analyses based only on QRS duration are incomplete in nature. Both QRS morphology and duration are important for patient selection as indicated by the guidelines, and we believe that changes in both—reflected in QRS area—should be considered when assessing ventricular electrical resynchronization. 

### 4.3. The Relevance of the LV Lead Position

The location where the transvenous LV lead is placed via the coronary sinus has been considered important for optimal effect of CRT. One of the earlier studies found that pacing from the lateral wall resulted in greater LVdP/dt_max_ increase than with anterior LV pacing [25]. Other studies have shown conflicting results regarding the LV lateral wall as the optimal pacing site [26,27,28]. A retrospective study of 457 consecutive patients implanted with a CRT device found that patients with a final LV lateral wall lead position failed to show significant 4-year survival benefit over patients with a final non-lateral LV pacing site [29]. As proposed recently by Wouters et al. [30], findings from the present study indicate that the optimal anatomical LV pacing location does not exist, but rather that the final anatomical location of the LV lead should be individualized.

In the present study, we distinguished between five different anatomical locations: anterior, anterolateral, lateral, posterolateral, and posterior. Most patients achieved the largest ∆QRS area and %∆LVdP/dt_max_ with a lateral and anterolateral LV pacing position. Interestingly, in this study, however, QRS area reduction was largest in two patients with anterior LV pacing and in one patient with posterior LV pacing, while acute hemodynamic change was largest with posterior pacing in one patient.

Pacing in or nearby scar tissue of patients with ischemic cardiomyopathy is an important example of unfavorable LV lead placement as it leads to worse resynchronization and outcome than pacing in viable tissue [10]. Compared to patients with a non-ischemic cardiomyopathy, CRT in patients with an ischemic cardiomyopathy led to lower reduction in QRS area, which in turn was associated with poor outcome [8]. The significant correlation between ∆QRS area and %∆LVdP/dt_max_ indicates that QRS area can be used to individualize optimal LV lead placement and choice of most appropriate electrode on the quadripolar LV pacing lead.

### 4.4. Clinical Implications and Future Perspectives

QRS area has great potential for application in routine clinical care. Currently, its calculation requires a standard digital ECG and computer software to transform the ECG into VCG, resulting in an objective and accurate electrocardiographic measurement [23]. Software for automatic QRS area calculation from standard 12-lead ECGs is not yet commercially available in current clinical practice. However, similar to electrocardiographic parameters such as PR-interval and QRS duration, QRS area can also be calculated and displayed on the ECG when software becomes commercially available in the future. In addition to the current analysis on pacing location, the relation between ΔQRS area and acute hemodynamic improvement could be studied with various AV- and VV-delays in future studies.

The currently best optimization methods available are the ECG and echocardiograph. As we can conclude from aforementioned studies, there is currently no evident proof that the investigated ECG markers including QRS duration can be used for CRT optimization [20,21], or for LV lead placement for that matter. As for echocardiography—whereas its ability to detail mechanical dyssynchrony and hemodynamic surrogates provide clear potential for echocardiographic parameters in CRT optimization—studies show conflicting results and its use for CRT optimization still needs to be investigated [31]. In addition, the use of echocardiography for LV lead placement during device implantation would be impractical.

### 4.5. Limitations

This study assessed the relative increase in LVdP/dt_max_ due to biventricular pacing as a marker for acute hemodynamic improvement, which has its limitations when translating the results to clinical or echocardiographic outcomes. Evidence for a relationship between acute hemodynamic response and long-term outcome is conflicting [32,33,34,35,36]. An explanation is that part of the therapy of resynchronization is to induce reverse remodelling, which is achieved in the long term by processes involving more than hemodynamic ones only [37].

This study has all the limitations that come with an observational cohort study. Furthermore, we were not able to systematically evaluate all pacing sites in each patient, because in the original studies the ECG and LVdP/dt_max_ measurements were not performed on all pacing sites. However, the findings from this study that QRS area reduction relates to acute hemodynamic response are in line with our previous finding that QRS area reduction is predictive of clinical and echocardiographic outcomes [8].

## 5. Conclusions

The present study provides evidence that the reduction in QRS area induced by biventricular pacing can be used to guide LV lead placement to its optimal pacing site. Within patients, a significant correlation was observed between ∆QRS area and ∆LVdP/dt_max_. In the vast majority of patients, the largest QRS area reduction coincided with the largest hemodynamic improvement.

## Figures and Tables

**Figure 1 jcm-10-05935-f001:**
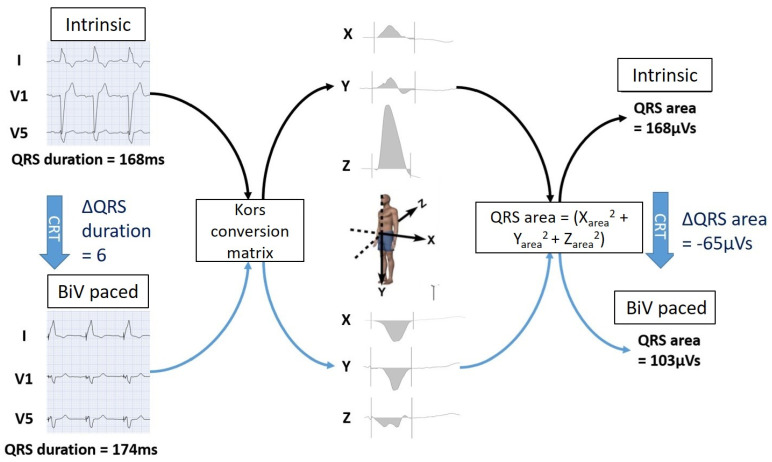
Transformation of ECG to VCG and calculation of (∆) QRS area. The 12-lead ECGs are mathematically converted in VCGs with the three orthogonal X-, Y-, and Z-leads using the Kors matrix. The X-, Y- and Z-leads of a patient with intrinsic rhythm and with CRT-pacing are shown. QRS area is calculated from the three orthogonal leads using the formula presented. As shown in this example, QRS area does not necessarily correlate with QRS duration. VCG = vectorcardiogram; CRT = cardiac resynchronization therapy. BiV pacing = biventricular pacing. Reprinted with permission from [8] Copyright © 2021 The Authors.

**Figure 2 jcm-10-05935-f002:**
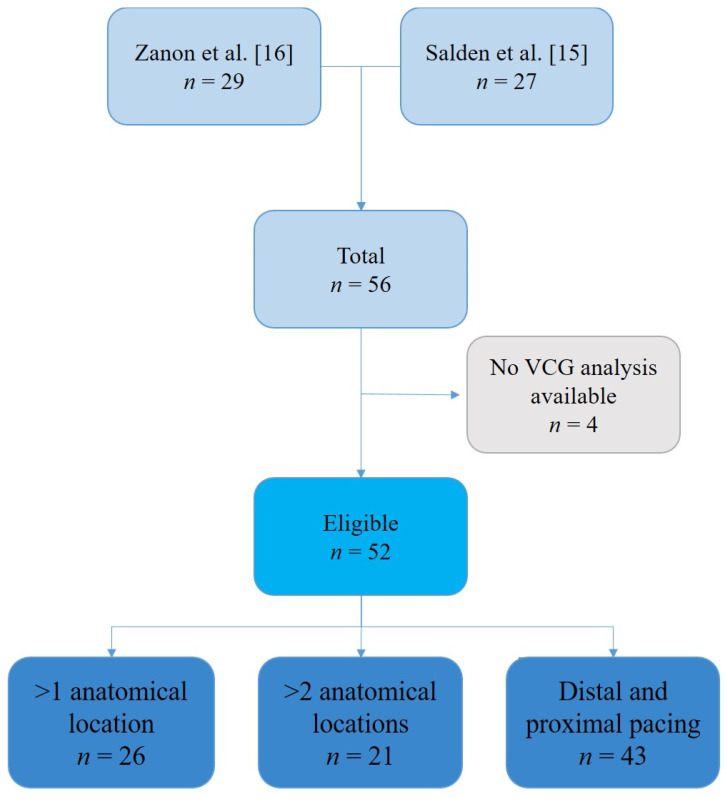
Flowchart of patients. A total of 56 patients were analyzed in previous studies. Four patients in which VCG analysis was not available were excluded, bringing the number of eligible patients to 52. Patients were analyzed depending on the number of different intra-individual pacing measurements. VCG = vectorcardiogram.

**Figure 3 jcm-10-05935-f003:**
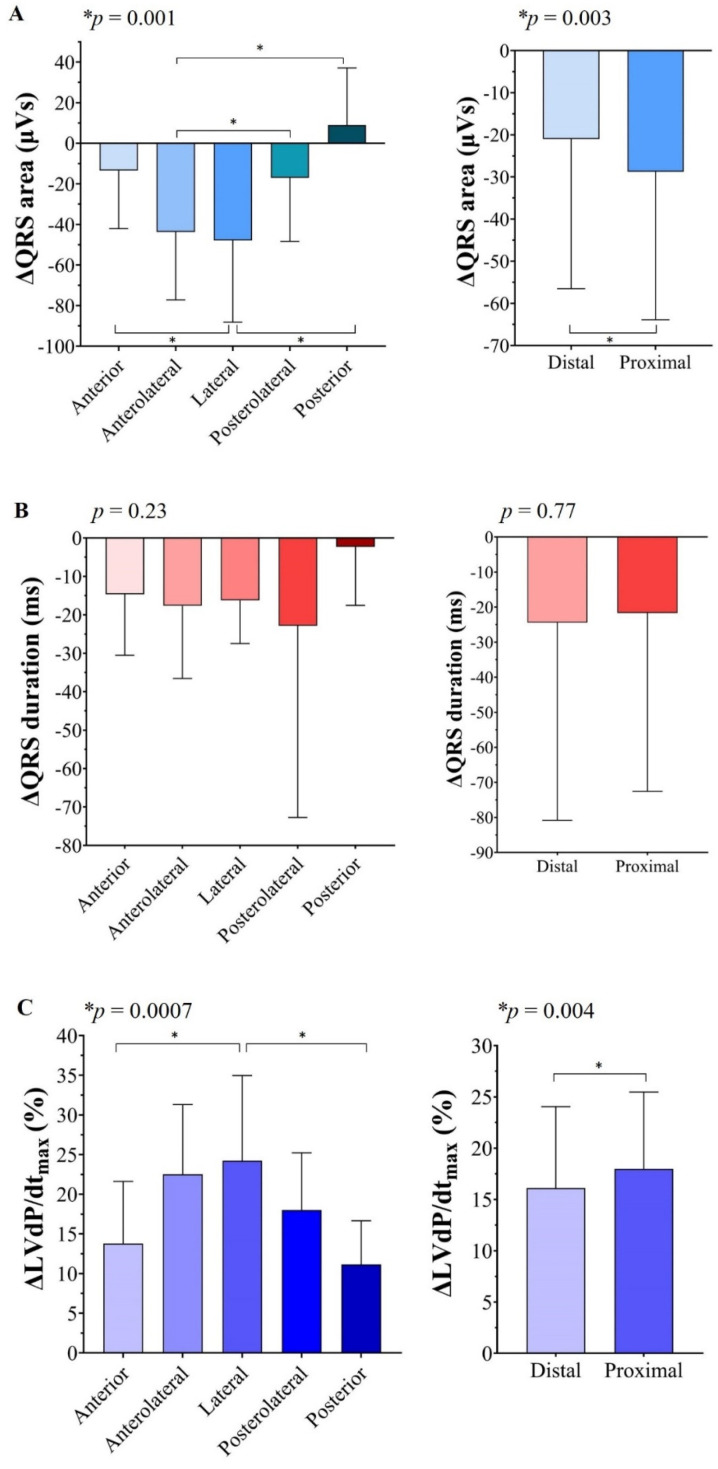
∆QRS area (**A**), ∆QRS duration (**B**), and ∆LVdP/dt_max_ (**C**) with different pacing locations. Left: *p*-values calculated with mixed-effects analysis and Sidak’s multiple comparison test. Right: *p*-values calculated with paired sample *T*-test. * *p* < 0.05.

**Figure 4 jcm-10-05935-f004:**
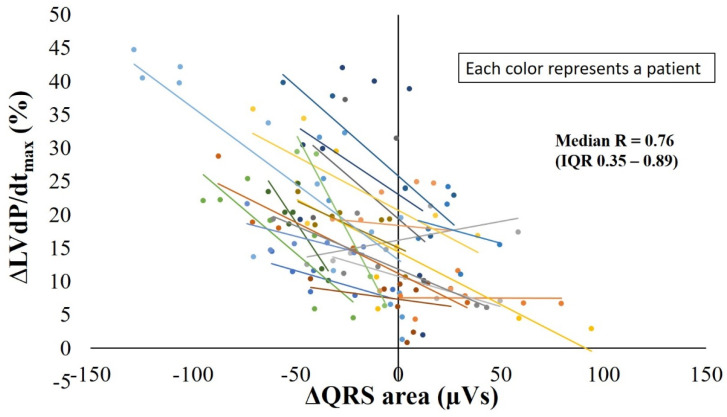
Plots of the relationship between ∆QRS area and ∆LVdP/dt_max_ in individual patients from Zanon et al. [16]. Only patients with at least three pacing measurement were plotted (*n* = 21). Median R and interquartile range (IQR) are displayed.

**Table 1 jcm-10-05935-t001:** Baseline characteristics of study population.

	All Patients (*n* = 52)
Age (y)	70 ± 9
Male (*n*, %)	41 (79)
NYHA	
II (*n*, %)	28 (54)
III (*n*, %)	24 (46)
Ischemic CMP (*n*, %)	26 (50)
Baseline LVEF (%)	29 ± 9
Sinus rhythm (*n*, %)	37 (71)
QRS duration (ms)	169 ± 18
LBBB (*n*, %)	35 (67)
IVCD (*n*, %)	6 (12)
RBBB (*n*, %)	4 (8)
Upgrade from RV pacing (*n*, %)	7 (13)
QRS area (µVs)	108 ± 44

NYHA = New York Heart Association; CMP = cardiomyopathy; LVEF = left ventricular ejection fraction; LBBB = left bundle branch block; IVCD = non-specific intraventricular conduction delay; RBBB = right bundle branch block; RV = right ventricular.

## Data Availability

The datasets generated during and/or analyzed during the current study are available from the corresponding author (M.G.) on reasonable request. The data are not publicly available due to their containing information that could compromise the privacy of research participants.

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
