# Peer review of "Left Ventricular Lead Placement Guided by Reduction in QRS Area"

_jcm, 2021, doi:10.3390/jcm10245935_

Round 1

Reviewer 1 Report

The authors of this manuscript investigate whether the reduction in QRS area is associated with hemodynamic improvement by pacing different LV sites. This is a very important issue, as for several years, different researchers have been studying the probability of response to cardiac resynchronization therapy.

the article is well written and as and presents original data.

in the caption of figure 1, it must be stated that the image is a reproduction of the original article (J Cardiovasc Electrophysiol. 2021;32(3):813-22.)

in figure 2 (flowchart of patients), the last square on the right side (distal pacing and proximal pacing) has a small typing error (ànd) that should be corrected.

I think the references are adequate and correct.

Author Response

I would like to thank the reviewer for this positive review.

1. Reviewer's comment: In the caption of figure 1, it must be stated that the image is a reproduction of the original article (J Cardiovasc Electrophysiol. 2021;32(3):813-22.)

Response: Thank you for this comment. I have added in the caption of figure 1:

"Image is reproduced from the original article (J Cardiovasc Electrophysiol. 2021;32(3):813-22.)"

2. Reviewer's comment: In figure 2 (flowchart of patients), the last square on the right side (distal pacing and proximal pacing) has a small typing error (ànd) that should be corrected.

Response: Thank you, the typo in figure 2 has been corrected.

Reviewer 2 Report

This was an excellent, small study from two centers investigating the change in QRS area for guidance of LV lead placement for CRT optimization in patients with heart failure. 

I would be interested to see what the results were stratified by LBBB and non-LBBB, as non-LBBB individuals are less likely to respond clinically. If a sub-analysis could be included, despite small number, it would increase the merits of the study. 

Is it possible to include a post-CRT implant ejection fraction to assess hemodynamic improvement by echocardiography?

Author Response

  1. I would be interested to see what the results were stratified by LBBB and non-LBBB, as non-LBBB individuals are less likely to respond clinically. If a sub-analysis could be included, despite small number, it would increase the merits of the study. 

Response: I would like to thank the reviewer for this positive review. We understand the question of this reviewer, however we think that stratification in subgroups would result in group sizes (in particular the one for the non-LBBB) that are too small to draw any conclusion from.

  1. Is it possible to include a post-CRT implant ejection fraction to assess hemodynamic improvement by echocardiography?

Response: Since both studies were set-up as acute hemodynamic studies to compare different LV pacing sites with the electrophysiological changes, we do not have the post-CRT echocardiographic data.

Reviewer 3 Report

Dear Authors,

I read article entitled 'Left ventricular lead placement guided by reduction in QRS area' with interest.

This original investigation concerns ΔQRS area in patients after CRT introduction and its associations with %ΔLVdP/dtmax at different LV pacing locations.

However, I have some minor comments regarding the paper:

  1. Please perform multiple linear regression analysis of predictors of %ΔLVdP/dtmax, including ΔQRS area, ΔQRS duration and clinical factors.
  2. Was there any difference in ΔQRS area in specific leads which could predict %ΔLVdP/dtmax at different LV pacing locations?
  3. How could these results be used in centres where the custom made Matlab software (Mathworks Inc.) is not available? Please discuss, especially in light of authors conclusion “QRS area, which is an easily, noninvasively obtainable and objective parameter, may be useful to guide LV lead placement in CRT”.
  4. Please show data with non-normal distribution as medians and interquartile ranges and compare the with appropriate non-parametric test.
  5. Figure 1 in the current paper is almost identical to Figure 1 in reference number 8. There is even no mention about this fact.
  6. Figure 2. Please correct a typo in the word “and”.
  7. Table 1. Please compare patients with ischaemic and non-ischaemic CMP, including ΔQRS area. P values comparing two groups should be shown in the last column.

Author Response

  1. Please perform multiple linear regression analysis of predictors of %ΔLVdP/dtmax, including ΔQRS area, ΔQRS duration and clinical factors.

Response: Thank you for this suggestion. A few issues arise when attempting a multivariable analysis. The main issue is the number of patients in this study, in which a maximum of four but mostly only two and even one parameter(s) can be tested at once. In addition, such multivariable regression analysis cannot be conducted with QRS area, LVdP/dtmax, and clinical characteristics as in this case it would combine both continuous and categorical variables.

  1. Was there any difference in ΔQRS area in specific leads which could predict %ΔLVdP/dtmax at different LV pacing locations?

Response: Thank you for this question. The correlations with %∆LVdP/dtmax were equally good for the 3D area (X-, Y-, and Z-lead) and Z-area. We chose to report for 3D area because this is consistent with our earlier work on QRS area.

  1. How could these results be used in centres where the custom made Matlab software (Mathworks Inc.) is not available? Please discuss, especially in light of authors conclusion “QRS area, which is an easily, noninvasively obtainable and objective parameter, may be useful to guide LV lead placement in CRT”.

Response: Currently the software for (automatic) QRS area calculation is indeed not commercially available, but potentially can become commercially available for e.g. ECG equipment which then can calculate QRS area in addition to displaying the ECG. I have added the following on page 10 under 4.4 “ Clinical implications and future perspective”:

Software for automatic QRS area calculation from standard 12-lead ECGs is not yet commercially available in current clinical practice. However, similar to electrocardiographic parameters such as PR-interval and QRS duration, QRS area can also be calculated and displayed on the ECG when software becomes commercially available in the future.”

  1. Please show data with non-normal distribution as medians and interquartile ranges and compare the with appropriate non-parametric test.

Response: Thank you for this comment. We have reassessed the outcome parameters (∆QRS area and ∆LVdP/dtmax) from the different LV lead pacing sites with multiple normality tests, and all passed the test for normality, except for the outcomes of the 21 patients used in figure 4 (hence, median + IQR was used there). Therefore, we think that displaying the average + standard deviation and correspondingly using the paired T-test would be more appropriate in this case.

  1. Figure 1 in the current paper is almost identical to Figure 1 in reference number 8. There is even no mention about this fact.

Response: Thank you for this comment. I have added in the caption below figure 1:

"Image is reproduced from the original article (J Cardiovasc Electrophysiol. 2021;32(3):813-22.)"

  1. Figure 2. Please correct a typo in the word “and”.

Response: Thank you, the typo in the figure has been corrected.

  1. Table 1. Please compare patients with ischaemic and non-ischaemic CMP, including ΔQRS area. P values comparing two groups should be shown in the last column.

Response: Thank you for this suggestion, however we think that dividing the group will result in group sizes that are too small to draw any conclusions from.

Reviewer 4 Report

In the paper "Left ventricular lead placement guided by reduction in QRS area", Ali Ghossein and coll. investigate whether the reduction in QRS area is associated with hemodynamic improvement by pacing different LV sites and can be used to guide LV lead placement. The paper is well written, the  research is well conducted and the results and the conclusion are clearly explained. 

Minor comments

-In order to better characterized the study population please add the table 1  HFrEF pharmacologic therapy of the patients enrolled in the study

-Please clarify how the ΔLVdP/dtmax  were evaluated in the Salden study   (the same of the Zanon study?)

- Please check the manuscript for typos and some grammatical errors.

Author Response

  1. In order to better characterized the study population please add the table 1  HFrEF pharmacologic therapy of the patients enrolled in the study

Response: We thank the reviewer for this important comment. Unfortunately, we only have the information on the pharmalogical treatment of 26 patients from the study of Salden et al., while the data for the patients from Zanon et al. could not be retrieved unfortunately. Although we very well understand this question of the reviewer, we strongly believe that this does not hamper the interpretation of the results.

  1. Please clarify how the ΔLVdP/dtmax  were evaluated in the Salden study   (the same of the Zanon study?)

Response: Thank you for this comment. I have added a description of how the hemodynamic changes were measured on page 3 under 2.2 “Study protocol and hemodynamic measurements:

“In both studies, the average LVdP/dtmax of at least 10 beats at baseline (AAI pacing) and during pacing from each LV lead position was measured in each patient. Ventricular extrasystoles were excluded. %∆LVdP/dtmax was calculated as the percentual change of LVdP/dtmax with every pacing site compared to baseline.”

  1. Please check the manuscript for typos and some grammatical errors.

Response: Thank you, I have revised the whole document and corrected some typos and grammatical errors using “track changes”.